# A 3D Unsupervised Domain Adaption Framework Combining Style Translation and Self-Training for Abdominal Organs Segmentation

Jiaxi Li[1][0009−0005−1936−6023], Qiang Chen[1][0009−0007−4870−1796], Haoyu Ding[1][0009−0004−1706−6660], Hongying Liu[1][0000−0001−5961−5569], and Liang Wan[1]✉[0000−0001−5501−9575]

Academy of Medical Engineering and Translational Medicine, Tianjin University, Tianjin 300072, China
lwan@tju.edu.cn

**Abstract.** Accurate segmentation of abdominal organs is crucial for the diagnosis and treatment of diseases. Thanks to the development of deep learning, the performance of CT abdominal organ segmentation has been qualitatively improved. However, due to the lack of labeled data for MR, it is challenging to utilize the existing CT data to achieve model adaptation on MR modality. Unsupervised domain adaptation has shown the potential to alleviate this challenge by learning from labeled source domain images as well as a large number of unlabeled target images. In this work, we first generate diverse fake MR data through a style translation network to assist in segmentation model training. Next, we follow a self-training strategy to utilize the segmentation network after training with mixed style images, and apply strategies such as pseudo-label filtering and elastic registration to generate accurate pseudo-labels for the MR data. Finally, we adopt a two-stage framework to localize the region of interest and then perform fine segmentation on it, which further improves the performance and efficiency of segmentation. Experiments on the validation set of FLARE 2024 demonstrate that our method achieves excellent segmentation performance as well as fast and low-resource model inference. The average DSC and NSD scores are 79.42% and 86.46%, respectively, the average inference time is 2.81 s, and the maximum GPU memory is 4135 MB on validation set. The code is available at https://github.com/TJUQiangChen/FLARE24-task3.

**Keywords:** Abdominal organs segmentation · Unsupervised domain adaption · Style translation · Self-training.

## 1 Introduction

Abdominal organs are primary sites for several prevalent cancers, with colorectal and pancreatic cancers ranking as the second and third leading causes of cancer-related deaths worldwide. As such, accurately segmenting abdominal organs is

vital for effective cancer diagnosis, treatment planning, and prognosis assessment. In recent years, deep convolutional neural networks have made significant strides in 3D medical image segmentation. Notably, the FLARE competitions from 2021 to 2023 provided extensive CT datasets along with some manually annotated results for both supervised and semi-supervised segmentation. This initiative has led to numerous outstanding contributions in the fields of CT abdominal organ segmentation [4,13]. However, despite these advancements, the segmentation of abdominal organs in MRI scans remains relatively underexplored. This gap primarily arises from the severe scarcity of labeled data for abdominal MRI within the research community; for instance, the training set in the MICCAI AMOS challenge only contains 40 labeled MRI scans [5]. Therefore, exploring how to utilize richly labeled CT image data to improve MR image segmentation becomes an important and challenging task.

Unsupervised domain adaptation (UDA) is a popular solution that aims to learn a given task model using annotated source and unlabeled target domains so that it performs well in the target domain to which the test dataset belongs. This approach enhances the adaptability of the model for real-world clinical applications and has become a focal point in medical image analysis research [14,17,11]. Cross-modality UDA, particularly between CT and MR images, presents unique challenges due to significant differences in data distribution. Most existing methods focus on image alignment, aiming to achieve cross-domain appearance conversion. For instance, Chartisias et al. [2] employed CycleGAN to generate target domain MR images from source domain CT images, subsequently training a separate segmentation network on these pseudo MR images. Cai et al. [1] enhanced CycleGAN with a shape consistency loss to better constrain the output of generator. Tomar et al. [12] utilized a learnable self-attention module to capture spatial semantic information and applied attention regularization loss to encourage orthogonality in attention maps, facilitating the transformation of distinct anatomical structures. However, these methods often face several limitations: (1) they usually generate an output of the average style of the target domain based on the source domain image. This reduces the diversity of the generated fake MR styles. (2) Due to the low contrast of MR images, the generated fake MR images often fail to capture semantic details of small structures, such as the gallbladder, esophagus, and adrenal glands, significantly impacting multi-class segmentation performance. (3) These approaches often overlook the intrinsic 3D information in medical images and instead utilize a 2D network, thereby missing out on crucial deep semantic information. Overall, these methods are suitable for tasks with less segmentation difficulty. In contrast, for multi-target multi-modality segmentation tasks with more difficult segmentation, the above problems degrade the overall performance of the target domain.

To address these challenges, we propose an unsupervised 3D cross-modality domain adaptation framework that integrates style translation and self-training, leveraging the large and diverse abdominal organ segmentation dataset provided by FLARE 2024. First, we realize the style transformation between CT and MR for one-to-many mappings by training a disentangle learning-based style trans-

lation network, which enhances the diversity of generated MR styles. Then, to fully utilize the MR unlabeled data, we combine the self-training strategy [18] with a segmentation network based on the joint training of fake MR and real CT to generate pseudo labels for the MR data. We then conduct an in-depth analysis of the MR dataset and implement strategies such as pseudo label filtering, iterative optimization, and elastic registration to effectively enhance the accuracy of the pseudo labels. Finally, to further improve efficiency of the model, we employ a two-stage framework [7] that initially localizes the region of interest followed by detailed fine segmentation. This approach leads to enhanced segmentation performance and allows for fast, resource-efficient model inference.

Our main contributions are as follows:

- We present a 3D UDA framework that combines style translation and self-training, which can effectively improve MR segmentation accuracy.

- We generate diverse fake MR images to facilitate segmentation network training based on style and semantic disentanglement and reconstruction.

- We apply the strategies of pseudo-label filtering, iterative optimization, and elastic registration to effectively improve the accuracy of pseudo-labels.

- We implement a two-stage framework that first localizes the region of interest and then performs fine segmentation, demonstrating strong performance and fast, low-resource inference on the FLARE 2024 validation set.

## 2 Method

We propose a cross-modality unsupervised adaptive abdominal organ segmentation method based on style translation and self-training, as shown in Fig 1. This method consists of three main steps: (1) training an image-to-image style translation network. To enhance the ability of model to adapt to the MR modality by simultaneously learning the style distributions of both CT and MR, we first perform semantic and style disentanglement by training Generative Adversarial Networks (GANs), followed by reorganizing the semantic features of CT with the style features of multi-modality MR to achieve CT-to-MR style translation. (2) Training the segmentation network using fake MR images and real CT images to generate pseudo-labels for the real MR images. We follow a self-training strategy to generate pseudo-labels for real MR images using a segmentation network based on fake MR trained with real CT. To increase the accuracy of these pseudo-labels, we employ techniques such as pseudo-label filtering, iterative optimization, and elastic alignment. (3) Utilizing both labeled and pseudo-labeled data to jointly train the segmentation network in the two-stage framework. Inspired by the third-place solution from FLARE 2022 [7], we adopt a two-stage framework in the third step, first based on a coarse segmentation model aiming at obtaining the rough location of the target organ from the whole MR volume. The fine segmentation model realizes the precise segmentation of the abdominal organs based on the ROI cropping of the coarse segmentation results. The above methods are described in detail in the following subsections.

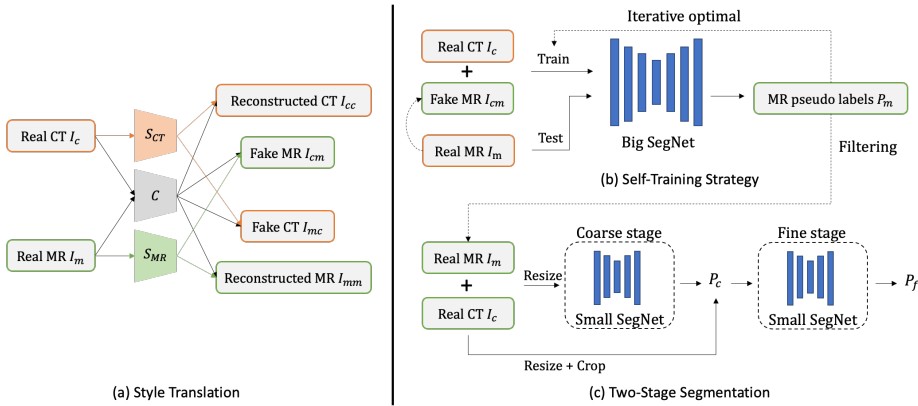

**Fig. 1.** Overview of our proposed cross-modality unsupervised domain adaptation abdominal organ segmentation method based on style translation and self-training.

### 2.1   Preprocessing

For the style translation network, we use the following preprocessing procedure:

- CT and MR were spatially normalized to a [4,1,1] spacing, where the ratio of the through-plane and in-plane resolution is around 4.
- All volumes are populated so that they have the same $512 \times 512$ pixel size in the XY plane.
- Applying min-max normalization to rescale the intensity to the range $[-1, 1]$.
- Dividing the 3D volumes into 2D slices.

For the segmentation network, we use the following preprocessing procedure:

- Image reorientation to the target direction.
- In the LLD-MMRI dataset [8], the C+Delay sequence with the best pseudo-labeling effect was used as a baseline to apply elastic alignment to images of other modalities of the same patient.
- For the Big SegNet, we resample images to uniform sizes [96, 256, 256].
- For the two-stage SegNet, We use small-scale images as the input to improve efficiency. Coarse input: [64, 64, 64]; Fine input: [96, 192, 192].
- We applied a z-score normalization based on the mean and standard deviation of the intensity values in the input volume.

### 2.2   Style Translation

To enable the model to learn both CT and MR style distributions for better adaptation to the target domain, we train a style translation network based on DARNet [16], as shown in Fig. 1(a). In our implementation, CT and multi-modality MR images are input into the style translation network. Following the

design of DARNet, the encoder decomposes each input image into content and style representations, reconstructing the image from the content features. The network is trained using pixel-level adversarial training to facilitate cross-domain image-to-image translation, while also incorporating content-level adversarial training to ensure effective content feature alignment with the shared content encoder, thus preserving semantic information. More details can be found in [16]. After training, we extract content from each CT image and random sample styles from MR dataset, allowing the CT dataset to reflect the style distribution of the MR dataset. To evaluate the effect of image alignment, we conduct visualization experiments on the CT-to-MR translation data across different modalities.

Figure 2 illustrates the conversion of CT images to MR images with various modality styles. As shown, the generated Fake MR images exhibit style similarities to the target MR modality styles, demonstrating both the plausibility and diversity of the generated image styles. Moreover, the Fake MR images maintain semantic structural consistency with the original CT images, and the internal organ sizes are identical, indicating that the style translation network successfully preserves crucial semantic structure information. Consequently, this style translation approach allows us to create a Fake MR dataset that corresponds to the CT dataset, which can be used to support the training of subsequent segmentation networks.

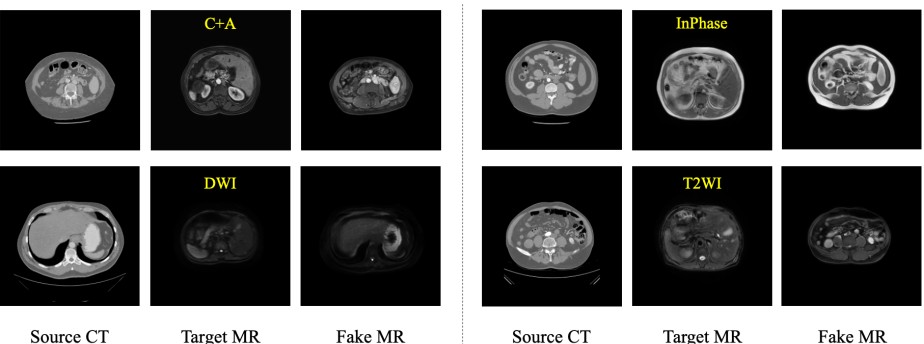

**Fig. 2.** CT-to-MR style translation visualization.

### 2.3   Pseudo Labeling

**Generation** To fully leverage the unlabeled MR data, it is crucial to generate accurate pseudo labels. We achieve this by using the Fake MR dataset and Real CT dataset, produced via style translation, for the model training of Big SegNet. The backbone network of Big SegNet is a segmentation network that combines convolutional layers with the Swin Transformer-PHTrans [7], which adopts a high-performance configuration with large model parameters and high computational capacity. After that, we can generate pseudo-labels for the Real MR dataset based on the trained Big SegNet.

However, for some special MR sequences, e.g., DWI, T2WI, they are poorly pseudo labeled, as shown in Figure 3. By examining the provided MR datasets, we observed that the LLD-MMRI dataset [8] comprises images of various sequence taken from the same patient, whereas the AMOS dataset contains multi sequences data from different patients. To leverage the unique characteristics of the LLD-MMRI dataset, we use the C+Delay sequence, in which the pseudo labeling has a high accuracy, as a benchmark for the registration of the images of other sequences. This makes it possible to reuse the pseudo labeling of the C+Delay sequence to other sequences. For the AMOS dataset, we not register and directly generate pseudo labels for its data. This approach significantly enhances the overall quality of pseudo labels in the LLD-MMRI dataset. The details of the registration are described in the next subsection.

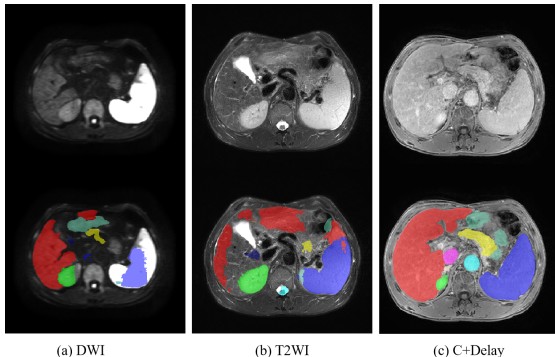

(a) DWI                    (b) T2WI                    (c) C+Delay

**Fig. 3.** A comparison of Big SegNet pseudo label generation across different sequences for the same patient. The first row showing the original images and the second row displaying the images with the pseudo labels.

**Elastic Registration** To register the 8 sequences for each patient in the LLD-MMRI dataset [8], we used the C+Delay sequence as a reference standard. We registered the remaining 7 sequences to this reference to ensure spatial alignment and achieve a uniform image size. The registration process was executed using the elastic registration function from the *ANTsPy* Python package. As shown in Figure 4, the registered InPhase sequence image aligns with the reference image in terms of spatial dimensions, allowing it to successfully reuse the pseudo-labels from the reference image.

It is important to note that some pseudo labels of the registered images and C+Delay sequence do not achieve full spatial alignment. As shown in Figure 5(a), the yellow box indicates the area of the image that should be the background, while the corresponding pseudo-label still has a value.Therefore, we apply a post-processing to the generated pseudo labels, setting the pixel values of these erroneous pseudo-labels to background values, as depicted in Fig. 5(b).

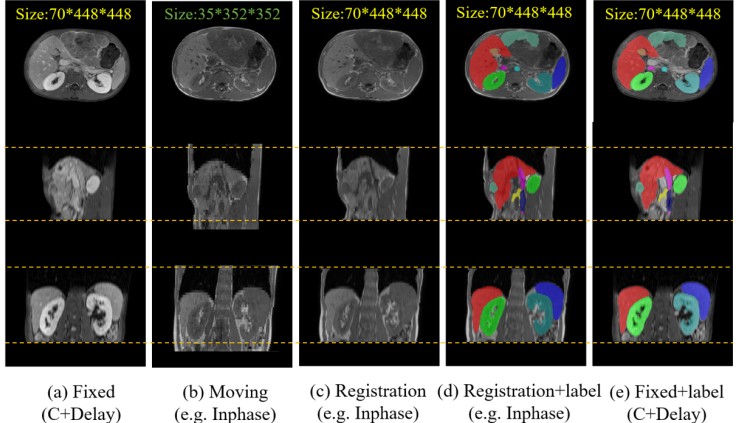

(a) Fixed
(C+Delay)
(b) Moving
(e.g. Inphase)
(c) Registration
(e.g. Inphase)
(d) Registration+label
(e.g. Inphase)
(e) Fixed+label
(C+Delay)

**Fig. 4.** Comparison images before and after registration. (a) Reference sequence C+Delay; (b) Image of the InPhase sequence to be aligned; (c) The image after registration; (d) The registered image and reused pseudo label (e) The reference image and its corresponding pseudo label. (The orange dashed line is the reference line, itk-snap visualizes the size of the image for reference only.)

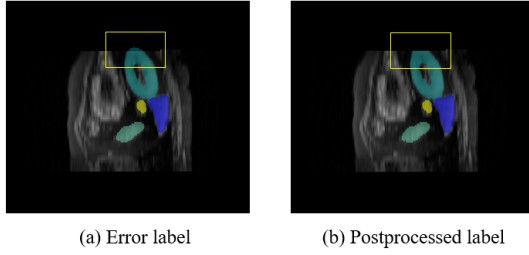

(a) Error label                    (b) Postprocessed label

**Fig. 5.** (a) Without post-processing, the pseudo-labels incorrectly assign values to the background regions. (b) After post-processing, the wrong labels are corrected.

**Filtering** To further improve the accuracy of pseudo labeling, we perform filtering on all generated pseudo-labels to prevent those with significant errors from being used in training. Leveraging the prior knowledge that organ volumes should be roughly similar in different adults, we filter out pseudo labels with organ volumes that are excessively large or small, as shown in Figure 6.

Specifically, we begin by calculating the average volume $V_{avg}^o$ for each organ category using the labeled CT dataset, where $o = \{1, \ldots, 13\}$ represents the total of 13 categories to be segmented. Next, we compute the average volume $V_p^o$ for each category based on the pseudo label. By comparing $V_p^o$ with the reference volumes $V_{avg}^o$, we can determine the reasonableness of the pseudo labels. The

process can be calculated as follows,

$$U = \prod_{o=1}^{13} \mathbb{I}[V_{avg}^o * l < V_p^o < V_{avg}^o * h], \tag{1}$$

where $\mathbb{I}(\cdot)$ is the indicator function, and $l$ and $h$ define the range of acceptable pseudo label volumes. When $U$ equals 1, it indicates that the physical volume of the pseudo label for all abdominal organs in a case falls within the specified range, allowing the pseudo-label to be used in subsequent training. Conversely, the pseudo-label is discarded. In our experiment, $l$ was set to 0.2 and $h$ to 1.8.

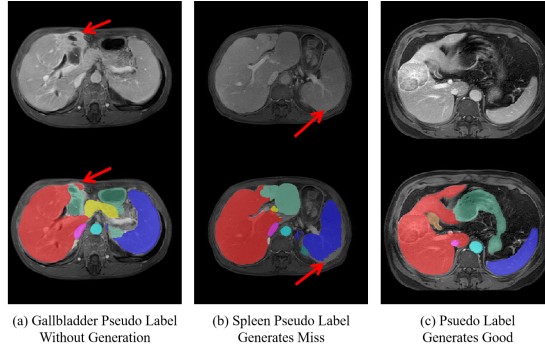



(a) Gallbladder Pseudo Label     (b) Spleen Pseudo Label     (c) Psuedo Label
Without Generation         Generates Miss         Generates Good



**Fig. 6.** The (a) and (b) pseudo labels were excluded by the filtering algorithm, while the pseudo label (c) was retained. **(a)** The arrow indicates the gallbladder pseudo label, which was incorrectly labeled as the stomach. **(b)** The arrow highlights a missing spleen pseudo label. **(c)** Demonstrates a well-generated pseudo label.

**Refine Pseudo Label** In addition, we iteratively optimize the generated pseudo labels to gradually enhance their quality. We first use the initially generated pseudo labels for the training of Real MR dataset and Real CT dataset. With the improved model resulting from this training, we then regenerate and filter the pseudo-labels to realize the iteration of pseudo labels.

### 2.4   Two-Stage Segmentation

**Segmentation Model** Our segmentation model (SegNet) adopts the same model structure as [7]. The model adopts a U-shaped encoder-decoder design, which is mainly composed of convolution modules and parallel hybrid modules. Among them, the parallel hybrid module comprises Transformer and CNN, which can model the hierarchical representation of local and global features. The symmetric decoder corresponding to the encoder is also built based on pure convolution and parallel hybrid modules. It fuses the semantic information of the encoder through skip connections and addition operations. The structure is simple and variable, and the number of Swin Transformer blocks and convolution blocks can be adjusted according to the medical image segmentation task. Please refer to [7] and the experimental details section for more details.

**Training** Inspired by the third-place solution of FLARE 2022 [7], we use a two-stage segmentation framework to enhance computational efficiency. The backbone network of the two-stage model is consistent with Big SegNet, but it has been designed with a reduced number of parameters. We introduced labeled CT and pseudo labeled MR together into the two-stage segmentation framework for training, as shown in Figure 7. Initially, a coarse segmentation model is trained, which aims to obtain the coarse location of the target organ from the whole data volume, i.e., to achieve the separation of foreground and background, and is a simple binary classification problem. Subsequently, the data is clipped based on the ROI obtained from the coarse segmentation, and input into Small SegNet for precise abdominal organ segmentation. It is important to note that the coarse and fine models do not share any network parameters.

**Inference** During the inference stage, we initially input the MR image into the coarse segmentation model to extract the foreground. Next, we implement test-time augmentation (TTA) along the anatomical axes (sagittal, coronal, and axial). This involves flipping the test image along these axes and feeding both the flipped and original test images into the fine segmentation model. We then average the predictions from both images to improve overall performance.

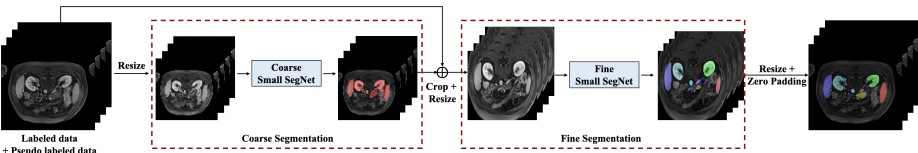

**Fig. 7.** Two-stage segmentation framework.

## 2.5 Post-processing

Post-processing based on connected components is commonly used in medical image segmentation. We eliminated false predictions in organ segmentation by retaining only the connected component with the largest prediction for each organ and removing all other components.

## 3 Experiments

### 3.1 Dataset and evaluation measures

The training dataset is curated from AMOS [5], LLD-MMRI [8], and past FLARE Challenges [9,10]. The training set includes 2050 abdomen CT scans and over 4000 MRI scans. For the CT dataset, we used the provided CT pseudo-labels

generated based on the FLARE 2022 winning algorithm [4]as training data. The validation and testing sets include 110 MRI scans from AMOS and 300 MRI scans of unknown source, respectively, which cover various MRI sequences, such as T1, T2, DWI, and so on.

The evaluation metrics encompass two accuracy measures—Dice Similarity Coefficient (DSC) and Normalized Surface Dice (NSD)—alongside two efficiency measures—running time and area under the GPU memory-time curve. These metrics collectively contribute to the ranking computation. Furthermore, the running time and GPU memory consumption are considered within tolerances of 15 seconds and 4 GB, respectively.

### 3.2   Implementation details

**Data augmentation** To alleviate the over-fitting of limited training data, we employed online data argumentation, including random rotation, scaling, adding white Gaussian noise, Gaussian blurring, adjusting rightness and contrast, simulation of low resolution, Gamma transformation, and elastic deformation.

**Training procedure** We use two different configurations of segmentation networks for pseudo-label generation and two-stage segmentation, respectively. To achieve high accuracy pseudo-label generation, Big SegNet uses a high performance configuration with large model parameters and high computational effort. We follow the hyper-parameter settings of [7], where the base number of channel C is 36 in Big SegNet, and the number of multi-head self-attentive heads used in different encoder stages is [3,6,12,24]. However, Small SegNet was configured as a lightweight architecture with faster training time and inference time, where the base number of channels is 16 and the number of multi-head self-attentive heads is [4,4,4,4]. The other model hyperparameter settings and the training scheme for the two-stage segmentation are shown in Table 1.

**Environments and requirements** The development environments and requirements are presented in Table 2.

## 4   Results and discussion

### 4.1   Quantitative results on validation set

Table 3 shows the final quantitative results on the public validation set. Our method achieves a mean DSC of 79.42% and a NSD of 86.46% on the FLARE 2024 public validation dataset.

---

[1] https://github.com/sksq96/pytorch-summary

[2] https://github.com/facebookresearch/fvcore

[3] https://github.com/lfwa/carbontracker

**Table 1.** Training protocols for the Big SegNet and two-stage samll SegNet.

| Model | Big SegNet / Small SegNet(coarse) / Small SegNet(fine) |
|---|---|
| Network initialization | "He" normal initialization |
| Batch size | 1 |
| Patch size | 96×256×256 / 64×64×64 / 96×192×192 |
| Total epochs | 300 |
| Optimizer | AdamW |
| Initial learning rate (lr) | 5e-4 |
| Lr decay schedule | Cosline Annealing LR |
| Training time | 24 / 2 / 16 hours |
| Loss function | Cross entropy + Dice |
| Number of model parameters | 6.6M / 4.2M / 4.2M [1] |
| Number of flops | 1037.77 / 18.60 / 251.19 G [2] |
| $CO_2$eq | 10.8251 / 0.0973 / 1.8788 Kg [3] |

**Table 2.** Development environments and requirements.

| System | Ubuntu 20.04.6 LTS |
|---|---|
| CPU | Intel(R) Xeon(R) Platinum 8153 CPU @ 2.00GHz |
| RAM | 192GB |
| GPU (number and type) | NVIDIA Tesla V100 32G |
| CUDA version | 11.3 |
| Programming language | Python 3.7.13 |
| Deep learning framework | torch 1.12, torchvision 0.13.0 |
| Specific dependencies | antspy, einops,fastremap,etc. |

To illustrate the rationality of the design of the modules, we conducted several experiments. Initially, we trained Big SegNet using labeled CT data to establish **baseline 1**, which achieved a DSC of only 39.02% on the MR validation set, highlighting difficulty of the task. We then included the generated Fake MR dataset to create **baseline 2**, which improved the DSC to 73.83%. This result demonstrates the effectiveness of style translation, as it significantly enhances cross-domain adaptation by enabling the model to learn the MR style distribution. Next, by using Real MR and Real CT data with iteratively optimized pseudo labels, we developed **baseline 3**, further increasing performance to 78.41%. This improvement shows that our strategies for pseudo label generation, filtering, and optimization effectively enhance label quality, aiding model training. Additionally, we experimented with Small SegNet, which has fewer parameters, to create **baseline 4**. This approach sped up inference but slightly reduced performance to a DSC of 77.63%. Finally, we verified the effectiveness of the two-stage segmentation framework by training Samll SegNet after the ROI of the abdominal organs, obtaining a **baseline 5** with a DSC of 79.19%

**Table 3.** Quantitative evaluation results.

| Target | Validation | |
|---|---|---|
| | DSC(%) | NSD(%) |
| Liver | 94.99 ± 2.00 | 95.48 ± 4.23 |
| Right kidney | 95.06 ± 2.28 | 94.08 ± 4.30 |
| Spleen | 94.95 ± 3.96 | 96.17 ± 5.95 |
| Pancreas | 77.90 ± 10.55 | 90.48 ± 9.37 |
| Aorta | 90.28 ± 8.82 | 94.07 ± 9.57 |
| Inferior vena cava | 84.32 ± 7.88 | 87.00 ± 9.41 |
| Right adrenal gland | 58.45 ± 14.18 | 75.97 ± 16.48 |
| Left adrenal gland | 61.40 ± 17.77 | 75.13 ± 21.70 |
| Gallbladder | 73.53 ± 25.50 | 67.42 ± 26.23 |
| Esophagus | 65.66 ± 12.66 | 83.16 ± 15.56 |
| Stomach | 79.62 ± 15.61 | 83.48 ± 17.14 |
| Duodenum | 61.06 ± 12.85 | 85.7 ± 11.55 |
| Left kidney | 95.18 ± 2.51 | 95.82 ± 3.76 |
| Average | 79.42 ± 13.72 | 86.46 ± 8.84 |

and faster inference. Incorporating TTA during the inference stage, we further boosted the final **Ours** DSC to 79.42%.

**Table 4.** Ablation Study On The Public Validation

| Baseline ID | Model | Training Data | | | Using Pseudo Label | Two-Stage | TTA | DSC(%) |
|---|---|---|---|---|---|---|---|---|
| | | CT(real) | MR(real) | MR(fake) | | | | |
| baseline 1 | Big SegNet | ✓ | | | | | | 39.02 |
| baseline 2 | Big SegNet | ✓ | | ✓ | | | | 73.83 |
| baseline 3 | Big SegNet | ✓ | ✓ | | ✓ | | | 78.41 |
| baseline 4 | Small SegNet | ✓ | ✓ | | ✓ | | | 77.63 |
| baseline 5 | Small SegNet | ✓ | ✓ | | ✓ | ✓ | | 79.19 |
| Ours | Small SegNet | ✓ | ✓ | | ✓ | ✓ | ✓ | **79.42** |

In addition, we verify the effectiveness of iterative optimization of pseudo labels, as shown in Table 5. The improvement of labeled iterative optimization gradually decreases, while the iterative optimization is time-consuming. Therefore, we finally use pseudo label V2 for training.

## 4.2  Qualitative results on validation set

We visualize the segmentation results of the validation set. According to the organizer's request, we show better examples in rows 1-2 and worse examples in rows 3-4. Representative samples in rows 1-2 of Figure 8(f) demonstrate our approach's effectiveness in capturing organ details. Thanks to the successful style translation and pseudo-labeling strategy, our method produces segmentation results closest to the ground truth compared to other baselines. As for the poorly

**Table 5.** The Results of Iterative Optimization Psuedo Label. Evaluation Metric: DSC(%)

| Model | Psuedo Label V1 | Psuedo Label V2 |
|---|---|---|
| Big SegNet | 78.35 | 78.41 |
| Small SetNet | 77.36 | 77.63 |

segmented lines 3-4, we analyze that it is due to the unclear boundaries of the MR image, while the example shown in line 4 is an image where the complete abdominal region is not intercepted, and this particular case is also one of the main reasons for the performance degradation. Additionally, the segmentation results improve progressively from left to right across each column. For instance, in the second row, the model initially detects only the entire left liver region (red). With the inclusion of pseudo-MR data and the pseudo-labeling strategy, the segmentation in columns (d) and (e) gradually improves, capturing additional organs like the pancreas (yellow), spleen (dark blue), and left kidney (dark green). However, the model was unable to segment some detailed organs, such as the gallbladder (brown), due to not applying the clipping ROI region in the two-stage segmentation. These visualization results indicate that our baseline can progressively enhance segmentation performance, and both the model and the strategy employed significantly contribute to this improvement.

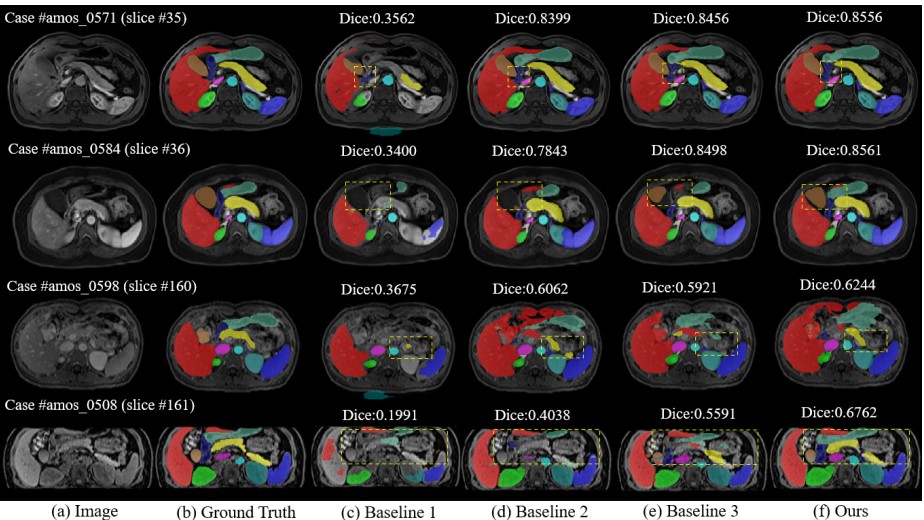

**Fig. 8.** Visualization of segmentation results of abdominal organs.

### 4.3   Segmentation efficiency results on validation set

The average running time in the validation set is 2.81 s per case in the inference phase, the average used GPU memory is 1919 MB and the max used GPU memory is 4135 MB. The area under the GPU memory-time curve is 584060. Table 6 lists the segmentation efficiency of some typical cases in running time including docker initialization time.

**Table 6.** Quantitative evaluation of segmentation efficiency in terms of the running time and GPU memory consumption. Total GPU denotes the area under the GPU Memory-Time curve. Evaluation GPU platform: NVIDIA Tesla V100 (32G).

| Case ID | Image Size | Running Time (s) | Max GPU (MB) | Total GPU (MB) |
|---------|------------|------------------|--------------|----------------|
| amos_0540 | (192, 192, 100) | 10.61 | 2859.12 | 10260.33 |
| amos_7324 | (256, 256, 80) | 10.69 | 3087.12 | 10409.21 |
| amos_0507 | (320, 290, 72) | 11.08 | 3009.12 | 10867.00 |
| amos_7236 | (400, 400, 115) | 11.42 | 3009.12 | 11468.00 |
| amos_7799 | (432, 432, 40) | 11.43 | 3087.12 | 11344.03 |
| amos_0557 | (512, 152, 512) | 14.10 | 2787.12 | 15283.26 |
| amos_0546 | (576, 468, 72) | 12.33 | 3009.12 | 12808.84 |
| amos_8082 | (1024, 1024, 82) | 18.03 | 3009.12 | 20945.54 |

### 4.4   Results on final testing set

We submitted the docker of our solution, which was evaluated by the challenge official on the test set, and the results are shown in tables 7 .

**Table 7.** The DSC, NSD, Running Time, and the area under the GPU memory-time curve on the test set from the official evaluation.

| DSC mean(%) | DSC median(%) | NSD mean(%) | NSD median(%) |
|-------------|---------------|-------------|---------------|
| 69.1 ± 16.6 | 73.6 | 71.3 ± 19.8 | 76.9 |
| Time mean(s) | Time median(s) | GPU mean(MB) | GPU median(MB) |
| 8.6 ± 0.4 | 8.5 | 477009.5 ± 26369.7 | 468786.2 |

### 4.5   Limitation and future work

The proposed method has several limitations. These limitations include: **1)** In terms of model backbone, some of the latest research advancements should try, such as SDSeg [6]. **2)** Since CT to MR style translation employs a 2D network, distortions may occur when applied to 3D data. **3)** The pseudo label selection

process did not undergo extensive exploration, and the current method for filtering pseudo labels is rather rudimentary. **4)** We plan to use TensorRT or TVM [3] to further accelerate inference and reduce GPU memory. In future work, efforts will be made to address these limitations to enhance model performance.

## 5   Conclusion

In this work, we developed a 3D unsupervised domain adaptation framework that integrates style translation and self-training to enhance MR segmentation accuracy. Using style translation, we generated a variety of fake MR target images to aid in training the segmentation network. We then incorporated a self-training strategy, alongside techniques such as pseudo-label filtering, iterative optimization, and elastic registration, to improve pseudo-label accuracy. Lastly, we implemented a two-stage framework for localizing regions of interest and conducting fine segmentation. This approach achieved strong segmentation performance and efficient inference on the FLARE 2024 validation set. In the future, we plan to optimize our style translation and self-training strategies to further enhance cross-domain segmentation performance and ensure fast, low-resource inference.

**Acknowledgements** Our implementation for participation in the FLARE 2024 challenge has neither used any pre-trained models nor additional datasets other than those provided by the organizers. The proposed solution is fully automatic without any manual intervention. We thank all data owners for making the CT scans publicly available and CodaLab  [15] for hosting the challenge platform.

## Disclosure of Interests

The authors declare no competing interests.

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

# 6    Response to reviewers

Thank you for allowing us to review and resubmit our manuscript, with the opportunity to address the comments of the reviewers. We are uploading (a) our point-by-point response to the comments (below) (response to reviewers) and (b) a clean updated manuscript with updated points.

**Reviewer#1, Concern#1:** In Table 1, the values in the "Number of model parameters" cell for Small SegNet (coarse) and (fine) are missing.

**Author response:** We have added the missing model parameters in Table 1.

**Reviewer#1, Concern#2:** In the section "2.3 Pseudo Labelling", "Labelling" is used, while the more common and correct term in this context is "Labeling".

**Author response:** We have corrected the misspelling of "labeling".

**Reviewer#1, Concern#3:** Please rephrase the caption of Fig. 5 and make it clear.

**Author response:** We have changed the title of Figure 5 to: (a) Without post-processing, the pseudo-labels incorrectly assign values to the background regions. (b) After post-processing, the wrong labels are corrected.

**Reviewer#2, Concern#1:** It would be better if the code was released.

**Author response:** We have made the code link public in the abstract section: https://github.com/TJUQiangChen/FLARE24-task3

**Reviewer#3, Concern#1:** However, there are some minor weaknesses in the paper. There are invalid superscripts in Table 1.

**Author response:** We have modified the footnote in Table 1 so that it can be linked normally.

**Reviewer#4, Concern#1:** Registration plays a crucial role in generating pseudo labels; what are the quantitative impacts of this process?

**Author response:** Registration plays an important role in the method. After experimental verification, we used unregistered data (directly using the pseudo labels generated by the model) to train Big SegNet and obtained an average DSC of 74.04%, while using the registered data for training, we obtained an average DSC of 78.41%. This proves that through registration between different modalities, pseudo labels generated in a better modality can be used on data in a modality with poor pseudo labels, making full use of multi-modal data training, thereby improving the segmentation results.

**Reviewer#4, Concern#2:** The architecture of the large and small SegNet models requires a more detailed description.

**Author response:** The structure of SegNet is consistent with the reference paper [7]. We provide a more detailed structural description of large and small SegNet models in the **Two-Stage Segmentation** section.

**Reviewer#4, Concern#3:** What specific settings were used for the MR dataset?

**Author response:** We do not have any special settings for the MR dataset. We use multiple preprocessing steps, which are described in Section 2.1 of the paper.

**Table 8.** Checklist Table. Please fill out this checklist table in the answer column.

| Requirements | Answer |
|---|---|
| A meaningful title | Yes |
| The number of authors ($\leq$6) | 5 |
| Author affiliations and ORCID | Yes |
| Corresponding author email is presented | Yes |
| Validation scores are presented in the abstract | Yes |
| Introduction includes at least three parts: background, related work, and motivation | Yes |
| A pipeline/network figure is provided | 1 |
| Pre-processing | 4 |
| Strategies to use the partial label | 4-5 |
| Strategies to use the unlabeled images. | 5-9 |
| Strategies to improve model inference | 9 |
| Post-processing | 9 |
| The dataset and evaluation metric section are presented | 9-10 |
| Environment setting table is provided | 2 |
| Training protocol table is provided | 1 |
| Ablation study | 11-12 |
| Efficiency evaluation results are provided | 11 |
| Visualized segmentation example is provided | 8 |
| Limitation and future work are presented | Yes |
| Reference format is consistent. | Yes |