# OpenReview forum: "A 3D Unsupervised Domain Adaption Framework Combining Style Translation and Self-Training for Abdominal Organs Segmentation"
_MICCAI.org/2024/Challenge/FLARE — FLARE 2024 withMinorRevisions_

### Official Review · Reviewer_MEns · 2025-01-24
**Comments**

**Rating:** 9
**Confidence:** 5

**Review:**

The authors introduce a novel framework for unsupervised segmentation, incorporating three key components: (1) style-content disentangled image translation, (2) pseudo-label filtering, iterative optimization, and elastic registration, and (3) a coarse-to-fine two-stage segmentation approach. By integrating these advanced technologies, the method achieves superior performance and presents an effective solution for unsupervised domain adaptation (UDA) tasks.

However, certain aspects of the framework warrant further clarification:

1.Registration plays a crucial role in generating pseudo labels; what are the quantitative impacts of this process?

2.The architecture of the large and small SegNet models requires more detailed description.

3.What specific settings were used for the MR dataset?

---

### Official Review · Reviewer_9RFg · 2025-01-25
**Review of "A 3D Unsupervised Domain Adaption Framework Combining Style Translation and Self-Training for Abdominal Organs Segmentation"**

**Rating:** 9
**Confidence:** 5

**Review:**

The authors propose a 3D UDA framework that combines style translation and self-training. It consists of three components:
(1)Generation of fake MR images based on disentanglement and reconstruction of style and semantics.
(2)Filtering pseudo-labels based on organ volume size.
(3)A coarse-to-fine two-stage segmentation framework.

However, there are some minor weaknesses in the paper.
There are invalid superscripts in Table 1.

---

### Official Review · Reviewer_DN6z · 2025-01-28
**comments**

**Rating:** 9
**Confidence:** 5

**Review:**

The authors proposed a novel 3D unsupervised domain adaptation framework that includes three key components: 1) style and semantic disentanglement and reconstruction for generating fake MR images. 2) pseudo-label filtering, iterative optimization, and elastic registration. 3) a two-stage framework for joint training.
However, there are some minor concerns in the paper.
1. It would be better if the code was released.

---

### Official Review · Reviewer_2QTU · 2025-03-04
**Review of "A 3D Unsupervised Domain Adaption Framework Combining Style Translation and Self-Training for Abdominal Organs Segmentation"**

**Rating:** 9
**Confidence:** 5

**Review:**

This paper presents a 3D unsupervised domain adaptation framework for abdominal organ segmentation, integrating style translation and self-training. By generating diverse fake MR data, it enhances the segmentation network training. The self-training strategy, along with pseudo-label filtering and elastic registration, improves pseudo-label accuracy. The two-stage framework also boosts segmentation performance and inference efficiency. The comments are listed below:

(1) In Table 1, the values in the "Number of model parameters" cell for Small SegNet (coarse) and (fine) are missing.
(2) In the section "2.3 Pseudo Labelling", "Labelling" is used, while the more common and correct term in this context is "Labeling".
(3) Please rephrase the caption of Fig. 5 and make it clear.

---

### Decision · Program_Chairs · 2025-03-20

**Decision:**

Accept

**Comment:**

Please carefully address the reviewers' comments in the revision.